# Thermal engineering of FAPbI$_3$ perovskite material via radiative thermal annealing and *in situ* XRD

Vanessa L. Pool[1,*], Benjia Dou[2,3,*], Douglas G. Van Campen[1], Talysa R. Klein-Stockert[2], Frank S. Barnes[3], Sean E. Shaheen[3,4], Md I. Ahmad[1,†], Maikel F.A.M. van Hest[2] & Michael F. Toney[1]

Lead halide perovskites have emerged as successful optoelectronic materials with high photovoltaic power conversion efficiencies and low material cost. However, substantial challenges remain in the scalability, stability and fundamental understanding of the materials. Here we present the application of radiative thermal annealing, an easily scalable processing method for synthesizing formamidinium lead iodide (FAPbI$_3$) perovskite solar absorbers. Devices fabricated from films formed via radiative thermal annealing have equivalent efficiencies to those annealed using a conventional hotplate. By coupling results from *in situ* X-ray diffraction using a radiative thermal annealing system with device performances, we mapped the processing phase space of FAPbI$_3$ and corresponding device efficiencies. Our map of processing-structure-performance space suggests the commonly used FAPbI$_3$ annealing time, 10 min at 170 °C, can be significantly reduced to 40 s at 170 °C without affecting the photovoltaic performance. The Johnson-Mehl-Avrami model was used to determine the activation energy for decomposition of FAPbI$_3$ into PbI$_2$.

[1] SLAC National Accelerator Laboratory, SSRL Materials Sciences Division, Menlo Park, California 94025, USA. [2] National Renewable Energy Laboratory (NREL), Materials Science Center, 15013 Denver West Parkway, Golden, Colorado 80401, USA. [3] Department of Electrical, Computer and Energy Engineering, University of Colorado Boulder, Boulder, Colorado 80309, USA. [4] Renewable and Sustainable Energy Institute, University of Colorado Boulder, Boulder, Colorado 80309, USA. † Present address: Indian Institute of Technology (BHU), Department of Ceramic Engineering, Varanasi 221005, India. * These authors contributed equally to this work. Correspondence and requests for materials should be addressed to M.F.A.M.v.H. (email: Maikel.van.Hest@nrel.gov).

In recent years, lead halide perovskite materials have attracted immense research interest due to their good charge transport, bandgap tunability, solution processability and excellent photovoltaic absorber properties. Reaching 22.1% (ref. 1) photovoltaic power conversion efficiency (PCE) within 6 years, the hybrid perovskites are unprecedented in the history of solar cell research. Recent intense compositional engineering works[2,3] further show the efficiency potential for these easily processed perovskite materials. In addition to high-efficiency solar cell applications, the materials have been studied for light-emitting diodes[4], lasers[5] and photodetectors[6].

To further improve perovskite film crystallinity and morphology in the perspective of processing, and thus to enhance the optoelectronic properties of the materials, research is mainly focused on three engineering approaches and their combinations. First, solvent/antisolvent engineering[7] that uses various solvents such as dimethylformamide (DMF), γ-butyrolactone and dimethylsulfoxide and antisolvents such as toluene, diethyl ether and chlorobenzene, which could dissolve the perovskite precursor solvent but do not dissolve the lead halide perovskites; second, intermediate engineering[8], which controls perovskite self-assembly crystallization process through forming certain intermediate state such as lead iodide ($PbI_2$) (dimethylsulfoxide); and last but not least, thermal annealing engineering[9], which explores a temperature induced perovskite phase transformation. Among these engineering methods, thermal annealing is the most widely studied processing method due to its simplicity and effectiveness. Various annealing conditions including maximum temperature[10], environment[11] and temperature profile[12] have been explored for forming perovskite materials. The importance of thermal annealing conditions is further amplified by the fact that temperature is one of the main drivers for perovskite decomposition[13]. However, so far, most of the thermal annealing has been performed on hotplates, and the annealing time is typically more than 5 min (refs 7,8) and times as long as 2 h (ref. 14) were reported. In the case of formamidinium lead triiodide ($FAPbI_3$), which is attracting increasing interests due to its higher thermal stability and broader optical absorption[8] (and the perovskite material used in this study) the standard annealing profile is 10 min at 170 °C on a hotplate[15,16]. The non-scalability and long processing time of the hotplate anneal made it not practical for large-scale production. For example, in roll-to-roll processing at $1 \, m \, s^{-1}$, a 10 min annealing would require a 600 m-long furnace, which is impractical for manufacturing.

Beyond hotplate annealing, there are a few reported studies on using optical annealing approaches. Troughton et al.[17,18] reported using near-infrared radiation (halogen lamp) and photonic flashing (xenon lamp), and Lavery et al.[19] proposed the use of intense pulsed light (xenon lamp) for sintering lead halide perovskites. The use of such optical annealing not only allows the sample to be effectively heated by absorption in the active layer, but also by absorption in the fluorine doped $SnO_2$ (FTO) coated substrate[20]. However, the previous works only studied methylammonium based perovskite, such as methylammonium $PbI_2$ ($MAPbI_3$) and mixed halide as $MAPbI_{3-x}Cl_x$, and the power conversion efficiencies are mostly not as good as those obtained by hotplate/oven annealing. Moreover, with such flash annealing techniques, it is not straightforward to control the temperature accurately, and therefore they are not very well suited for conducting temperature-related studies that are important in the perovskite field.

In addition to the need of scalable and accurately controlled annealing techniques, fundamental understanding of the perovskite crystal structure formation, degradation, and phase transformation and its effect on the photovoltaic performance is key to further improve the stability and efficiency of perovskite solar cells. In situ X-ray diffraction (XRD) offers such insights, as evidenced by studies[21–25] performed on methylammonium-based perovskites. However, to the best of our knowledge, there is only one publication on in situ diffraction of $FAPbI_3$ based perovskites by Aguiar et al.[26] where they found 175 °C as the optimum processing temperature for $FAPbI_3$. This is consistent with the standard annealing profile of 10 min at 170 °C for the formation of $FAPbI_3$ (refs 15,16). It is noteworthy that solution-deposited $FAPbI_3$ forms a hexagonal precursor phase that transforms to the perovskite trigonal phase on annealing above 130 °C (refs 27,28). This transformation temperature is slightly higher than the bulk transition temperature of 125 °C (ref. 27). A better understanding of the crystal structure–cell efficiency relation will be beneficial to $FAPbI_3$-based perovskite solar cells.

This work presents the application of a radiative thermal annealing (RTA) to the $FAPbI_3$ perovskite system. Samples annealed using RTA show comparable efficiencies to samples made on a hotplate, which validates the use of in situ characterization using an RTA system for understanding the dynamics of the $FAPbI_3$ phases. Using an in situ XRD RTA system, this work effectively and efficiently monitors the temperature-induced phase transformation dynamics, primarily crystal structural transformation and degradation, in $FAPbI_3$ films. Based on RTA in situ XRD data and device performance data, we produced a processing structure performance space map that identified the time and temperature ranges that can be used to produce good quality perovskite films. These ranges are much broader than the standard $FAPbI_3$ annealing time, 10 min at 170 °C. For example, for any temperature between 170 and 210 °C, annealing times as low as 40 s can be used without affecting the photovoltaic performance. This result and the application for a RTA method will make processing of $FAPbI_3$ more scalable as the temperature profile is comparable to that of belt furnaces typically used in industrial manufacturing. In addition, the $FAPbI_3$ film decomposition process and activation energy are examined using established kinetic models, and a quantitative value of $FAPbI_3$ decomposition activation energy is obtained, which is potentially useful to determine the inherent lifetime of $FAPbI_3$.

## Results

**RTA of $FAPbI_3$ perovskite film**. RTA is widely used in the semiconductor industry due to its ease in achieving high temperatures and fast ramp rates. It is also cost effective and provides better temperature control and potential access to metastable states[29]. Supplementary Fig. 1 shows the cross-section of the RTA chamber used to produce the devices for this study. This system uses light from halogen lamps without any filter as the heating source to anneal samples with controlled radiation. To compare RTA with the more common conductive annealing in $FAPbI_3$ films, first the $FAPbI_3$ films were spin coated with a solvent engineering method[15], which involves using DMF as the solvent and toluene as the antisolvent (see Methods section for more details). The as spin-coated films were annealed with both hotplate and RTA for a variety of times and hold temperatures. The given annealing time is the sum of ramping time and hold time (Supplementary Fig. 2). The annealing ramp rate for the RTA was chosen as $10 °C \, s^{-1}$, to mimic the annealing ramping condition on a hotplate. We refer to the hotplate temperature as the set or hold temperature.

The ultraviolet-visible absorption spectra in Fig. 1a shows that the films made with the hotplate and RTA have an almost identical absorption spectra, which shows that $FAPbI_3$ films fabricated with these two methods have the same optical properties. Further comparison of the $FAPbI_3$ films made with a

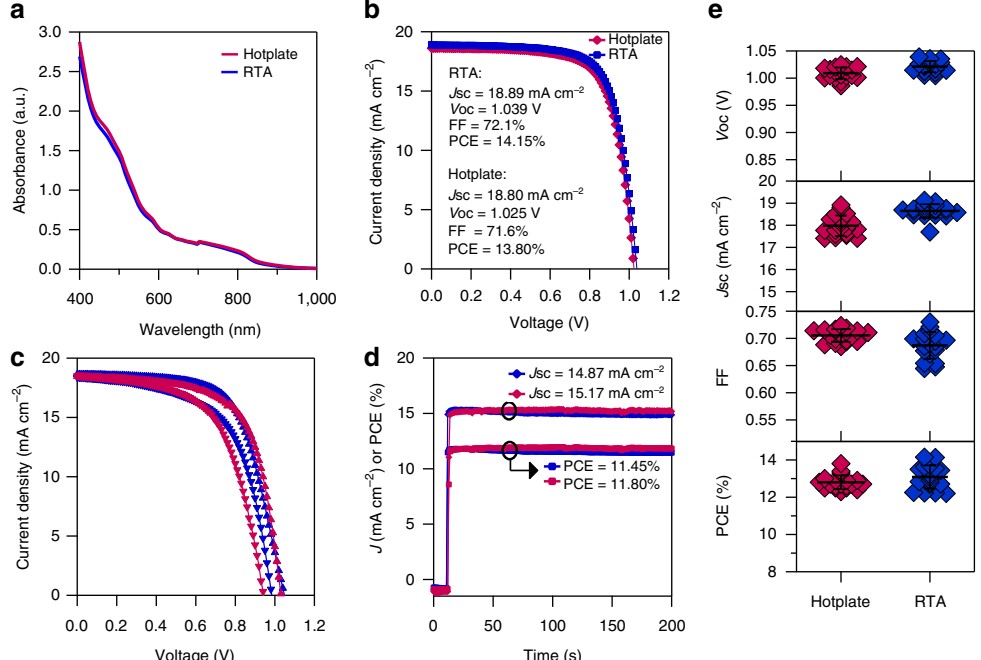

**Figure 1 | RTA comparison with hotplate for FAPbI₃ thermal processing.** (**a**) Ultraviolet-visible absorption spectra of FAPbI₃ films made with hotplate (red line) and RTA (blue line). (**b**) J–V curves for the best FAPbI₃ cells made with hotplate (red line) and RTA (blue line), measured by reverse (open circuit → short circuit) scan. (**c**) J–V curves for typical devices measured by forward (short circuit → open circuit) and reverse scans. (**d**) Stabilized photocurrent density and PCE at maximum power points for typical devices. (**e**) PV parameters for 19 cells of FAPbI₃ devices made with hotplate (red squares) and RTA (blue squares), and their average values (thick black line) and s.d. (thin black lines). All the perovskite films in this figure were annealed at 170 °C for 10 min.

hotplate and RTA was done by fabricating various planar structured devices with an architecture of FTO glass/compact-$TiO_2$/FAPbI₃/Spiro-OMeTAD/Au. Current density-voltage (J–V) curves, reverse scanned under one sun illumination, of the best planar devices made on a hotplate and in the RTA are presented in Fig. 1b. Both reverse and forward scanned J–V and stabilized PCE of typical devices are presented in Fig. 1c,d. As can be seen from the data shown in Fig. 1b–d, the photovoltaic performance of FAPbI₃ devices made with the RTA is equivalent to the device made using a hotplate. This was further confirmed with the photovoltaic performance data in Fig. 1e, where the open circuit voltage, short circuit current, fill factor and PCE, obtained by reversed J–V scan (complete scan conditions can be seen in the Methods section), of 19 devices using RTA and 19 devices using hotplate annealing are presented. These equivalent optical and device characterizations confirm that RTA can be used as a substitution for hotplate annealing. This also suggests that RTA does not induce a photo catalytic effect on the film. Therefore, the *in situ* studies performed below are relevant for perovskite films fabricated on hotplates.

**Rapid thermal annealing and *in situ* XRD characterization.** To fundamentally understand the FAPbI₃ film growth and decomposition process, the *in situ* XRD RTA system described in Ahmad *et al.*[29] was used, which allows measurements of the phase progression of FAPbI₃ during annealing. Using a ramp rate of 10 °C s⁻¹, which slows to ~5 °C s⁻¹ near the hold temperature to prevent overshoot, anneals were done for a number of temperatures around the standard temperature of 170 °C to investigate the conversion to the FAPbI₃ perovskite phase and the degradation of FAPbI₃ to $PbI_2$. Owing to the X-ray-sensitive nature of the samples, the total X-ray exposure was

limited to a total of 30 s per anneal, which was determined to produce negligible film damage (see Methods section for more details). For higher temperatures where the entire run was ~60 s, a single set of 60 scans with a 0.5 s exposure time was used. For longer anneal times (lower temperatures), two sets of 30 scans with different time spacing between scans were used, again with exposure time of 0.5 s (see Methods section for more details). This allows for monitoring of the progression of the material from the precursor to the perovskite with sufficient temporal resolution for the hexagonal–trigonal transition as well as the degradation from perovskite into $PbI_2$.

Figure 2a shows the *in situ* XRD data from a sample heated to 330 °C at 10 °C s⁻¹. This shows the progression of phases formed during annealing, and three different phases are observed. The sample starts out in a precursor phase, which is FAPbI₃ in the hexagonal $P6_3mc$ space group, a non-perovskite polymorph of the desired perovskite phase[27,28]. Upon heating to a temperature above 130 °C, the film converts to the trigonal $P3m1$ phase, the desired FAPbI₃ perovskite phase. When annealed for long enough and/or to high-enough temperature, the sample degrades to $PbI_2$ ($P\bar{3}m1$ phase). All three of the phases are identified and compared with the measured XRD in Fig. 2b. Full peak identification is included in Supplementary Fig. 3. In this work, the peak area of one or more peaks for each phase is utilized to track the phase conversions observed during the annealing. These chosen peaks are identified by the dashed boxes in Fig. 2a. For the hexagonal FAPbI₃ phase, the (2$\bar{1}$0) peak ($Q = 1.44$ A⁻¹) is tracked; for the trigonal FAPbI₃ phase, the overlapping (20$\bar{1}$) and (102) peaks ($Q = 1.39$ A⁻¹) are tracked; and for the $PbI_2$ phase, the overlapping (10$\bar{1}$), (101) and (002) peaks ($Q = 1.79$ A⁻¹) are tracked. Figure 2c–e shows three representative samples with set temperatures of 130, 170 and 330 °C, respectively, spanning the temperature range used. The

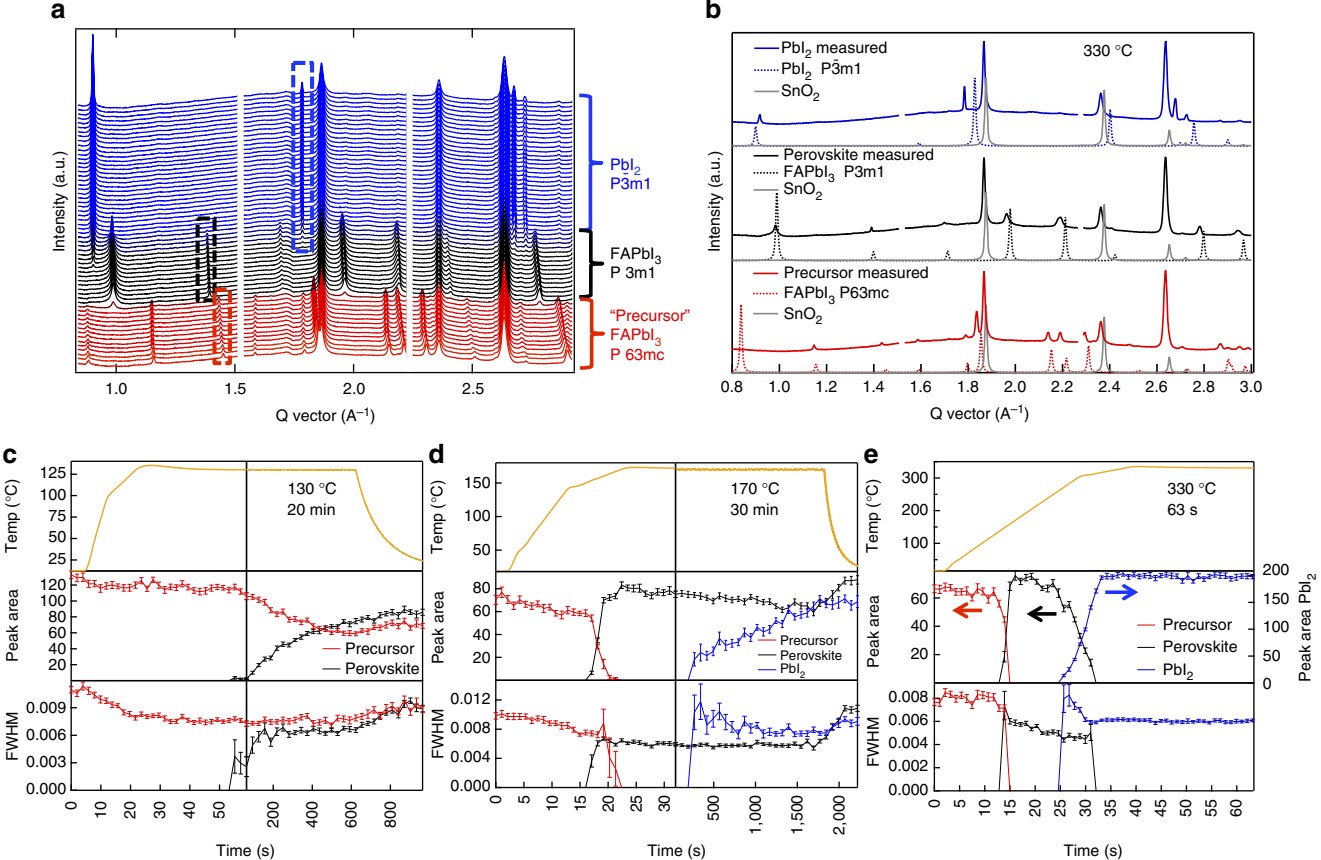

**Figure 2 | RTA/*in situ* XRD.** (**a**) *In situ* diffraction from a 330 °C sample with the phase progression indicated. (**b**) Phase identification from *in situ* 330 °C sample. From top to bottom the scans were taken at times 10.7, 16.1 and 42.9 s. (**c–e**) Integrated intensities of peaks from the precursor (red lines), perovskite (black lines) and PbI₂ (blue lines) phases as a function of time for films annealed at 130, 170 and 330 °C, respectively. The error bars are s.d.

temperature profile, peak area and full width half maximum (FWHM) are plotted versus time allowing the structural transformation and crystallite size to be tracked, and correlated to the temperature profile.

The transition from the precursor to the FAPbI₃ perovskite occurs rapidly (within 2 s), shown for 170 °C (Fig. 2d) and 330 °C (Fig. 2e), and is fast unless the temperature is near the transition temperature of ~130 °C, as seen in Fig. 2c. This is expected given that the phase transition from the hexagonal polymorph to the trigonal FAPbI₃ perovskite is at ~130 °C (see Fig. 2d,e). This transition temperature is consistent with previous studies by Hanusch *et al.*[30] and Stoumpos *et al.*[28].

Over the range of set temperatures, a coexistence of hexagonal and trigonal FAPbI₃ is observed, which is likely to be a result the polycrystalline and heterogeneous nature of the film with some grains/regions converting more readily than others. This is particularly apparent in the case of the 130 °C anneal (Fig. 2c), where not all of the film converts even after 20 min at temperature. During this transition, Fig. 2c–e shows that there is no change in the XRD peak FWHM, suggesting that the precursor (untransformed) and perovskite (transformed) crystallites are the same size. This behaviour suggests that in the nucleation and growth of the perovskite from the precursor perovskite, once nucleation takes place, the perovskite growth is fast: there is only one nuclei per precursor grain.

The slower conversion from the FAPbI₃ perovskite to PbI₂ probably reflects the fact that the formamidinium iodide (FAI) must leave the film for it to convert to PbI₂. Within

the time scale of our measurements (900 s), degradation of FAPbI₃ into PbI₂ is observed for films annealed to 170 °C or higher (Fig. 2d). For the sample heated to 330 °C, the film converted completely to PbI₂ within 33 s, which is before the sample reached the 330 °C set temperature. For all anneals done here, there is never direct conversion from the precursor films into PbI₂—the perovskite phase was always observed as an intermediate.

**Effect of annealing time on photovoltaic performance at 170 °C.** To confirm the lack of grain growth for samples after the transition into the perovskite phase, samples were annealed at 170 °C for 40 s and 10 min, and their morphology was compared using scanning electron microscopy (SEM; Fig. 3,b). For these two anneal times, the grain size is comparable, and the only noticeable difference is a small amount of PbI₂ seen by lighter spots on the sample annealed for 10 min (Fig. 3b). Although there is no substantial coarsening in the film at longer anneal times, there are other processes that could improve or degrade the device performance with additional annealing. To determine whether the phase purity alone is a direct metric for device performance, we made a set of devices with active FAPbI₃ layer annealed at 170 °C for 40 s, 100 s, 5 min and 10 min. The results of the photovoltaic performance are presented in Fig. 3c, showing that the difference in efficiency is within the error of the sample sets. Importantly, this demonstrates that the long annealing time (>40 s) is not necessary to make a high-efficiency device.

Here, it is worth noting that both the *in situ* XRD data and SEM images taken on perovskite films with 170 °C for 10 min show the existence of $PbI_2$, but the photovoltaic performance is not significantly different from devices made using a perovskite film showing no $PbI_2$. This indicates that a small amount of

$PbI_2$ in the perovskite film does not affect the initial performance of the $FAPbI_3$ devices. A similar observation has been made for the $MAPbI_3$ perovskite system[31–33]. The role of excess $PbI_2$ in perovskite films has been widely debated[31], and various mechanisms have been proposed to explain how some $PbI_2$ could be beneficial to the photovoltaic performance. Some possible mechanisms for this include that $PbI_2$ helps to form a favourable energy band alignment[32], or it may accumulate at grain boundaries and hinder charge recombination[33].

To further study how $FAPbI_3$ degradation affects the photovoltaic performance, devices were fabricated with the active layer annealed at 150, 190 and 210 °C, and their performance is presented in Supplementary Fig. 4. For all these, if the sample appeared to have converted to the perovskite and did not have visible color change, the efficiencies were very similar to that obtained at 170 °C, confirming the hypothesis that once the phase transition has occurred the efficiency is constant until significant degradation into $PbI_2$ occurs (see Supplementary Fig. 4c for device performance, and crystal information and see Supplementary Fig. 5 for SEM).

**Processing phase space**. To gain further insight into processing time and temperature, we have mapped a more comprehensive picture of processing phase space and corresponding device performance. To investigate the phase space of the $FAPbI_3$ formation and decomposition, the peak analysis of each of the samples (as shown in Fig. 2c–e) was used to determine the time and temperature at which a phase appears or disappears. The details of how this was determined are discussed in the *in situ* XRD data collection and analysis subsection of the Methods section. The conversion zone diagram is shown in Supplementary Fig. 6, which summarizes our temperature-time experiments by plotting the phases observed as a function of sample temperature at a given time during the annealing ramp. From Supplementary Fig. 6, zones of conversion are apparent where there are phase transformations, that is, conversion zone (from precursor to perovskite) and degradation zone (from perovskite to $PbI_2$).

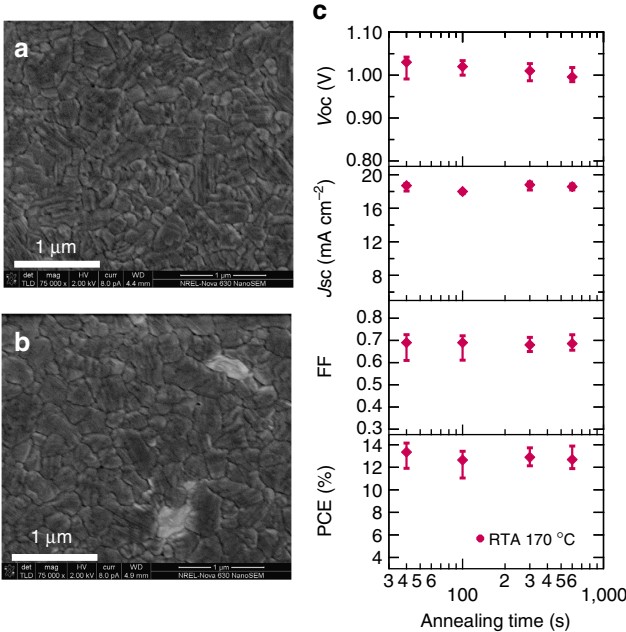

**Figure 3 | FAPbI₃ film morphology and device performance for film annealed at 170 °C.** (**a**) SEM image (scale bar, 1 μm) of FAPbI₃ film annealed at 170 °C for 40 s. (**b**) SEM image (scale bar, 1 μm) of FAPbI₃ film annealed at 170 °C for 5 min. (**c**) Annealing time dependence of photovoltaic performance (average, maximum and minimum values show by the square dots and error bars) for devices with active layer fabricated at 170 °C.

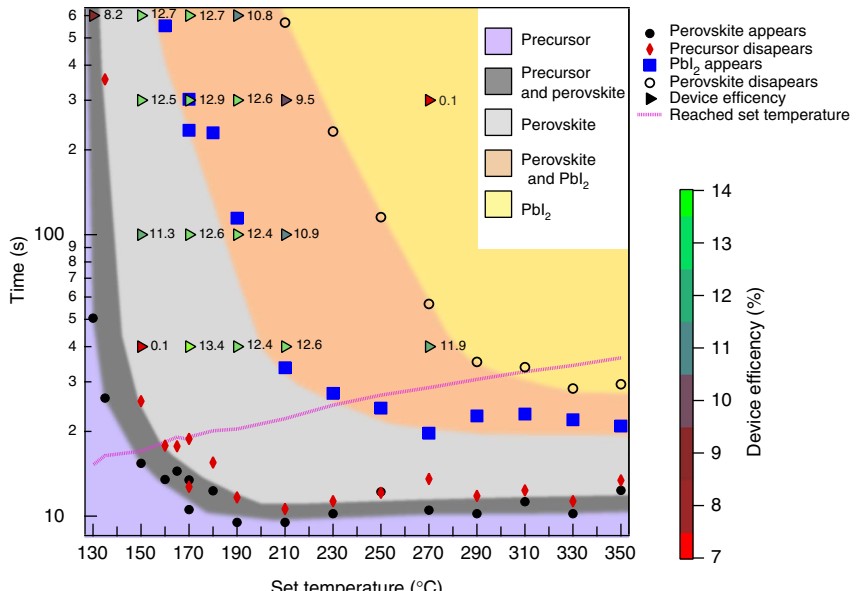

**Figure 4 | Processing phase space and relevant photovoltaic performances.** The film phase is determined from the perovskite appearance (solid black circles), the precursor disappearance (red diamonds), the PbI₂ appearance (blue squares) and the disappearance of the perovskite (open black circles). The time where the set temperature is reached in the annealing ramp is shown by a pink line. The efficiency data is represented by triangles.

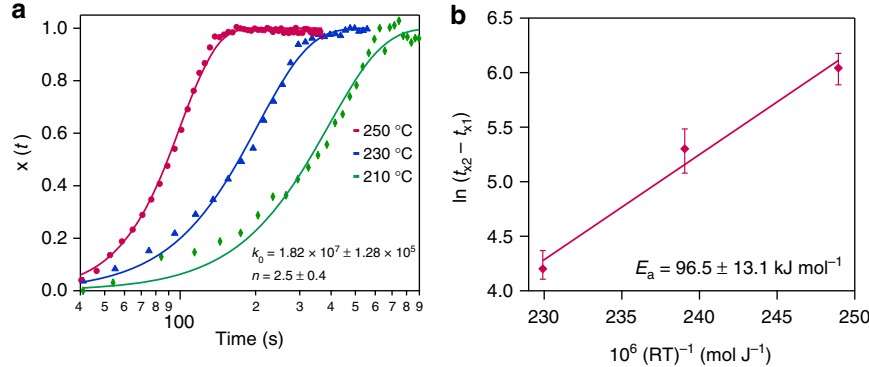

**Figure 5 | FAPbI₃ decomposition and kinetic modelling. (a)** Transformation fraction $x(t)$ as a function of time extracted from isothermal RTA with *in situ* XRD data at 210 °C (green, diamond), 230 °C (blue, triangle) and 250 °C (red, circle). Solid lines are fitting results with the JMA model (equation 2). **(b)** Reaction time versus inverse temperature from **a** by fitting equation 1. The error bars are the maximum and minimum values.

With Supplementary Fig. 6, we can calculate the processing phase space diagram. This is shown in Fig. 4 and plots the film phase as a function of annealing time verses set temperature for samples processed using RTA. The different coloured regions show the time-set temperature where the hexagonal FAPbI₃ (precursor), trigonal FAPbI₃ (perovskite) and PbI₂ form, as well as regions of phase coexistence. The hotplate annealed version of this figure is plotted in Supplementary Fig. 7. Presenting the data this way makes it possible to overlay the device efficiency data on the same graph as the crystal phase data. Efficiency data for samples made with equivalent temperature profiles was measured for a set of 18 sample conditions, and the average efficiency is plotted for each condition in Fig. 4. From Fig. 4 it is possible to identify optimal anneal times for a given set temperature to maximize the FAPbI₃ cell efficiency. Vertical cuts in Fig. 4 represent the phase evolution with time for a given set temperature (with a ramp rate of $10 °C s^{-1}$) and so inform the optimal time to anneal a sample to produce the desired crystal structure. The overlay of the device efficiency (given by the triangles) and the diffraction data shows that the samples with the best performance are in the region were the perovskite is fully formed (the light grey region) but a significant amount of PbI₂ has not yet formed (not too far into the orange region). The information in Fig. 4 thus informs further optimization of the annealing step in the perovskite synthesis. Not only is this useful for achieving good efficiency, but it also informs the use of lower anneal times reducing the thermal load and so reducing the cost of material manufacturing.

**FAPbI₃ decomposition kinetics.** Radiative annealing with *in situ* X-ray experiments and resulting processing space as discussed in the previous sections suggest FAPbI₃ decomposition to PbI₂ is a slow, thermally driven transformation. To better understand the kinetics of this transformation process, we quantitatively evaluate the experimental data with well-developed kinetic models[34]. Previously, Moore *et al.*[23] have studied crystallization kinetics of $CH_3NH_3PbX_3$, where $X$ represents lead salts such as chloride, iodide, nitride and acetate, and the proposed Johnson–Mehl–Avrami (JMA) modeling of $CH_3NH_3PbX_3$ crystallization can reveal important details of the perovskite formation.

Here, the kinetics of FAPbI₃ decomposition by applying the JMA model to our radiative annealing with *in situ* XRD data is explored. To mathematically perform a materials' phase transformation kinetic modelling, a reliable transformational fraction parameter needs to be identified and traced. The PbI₂ XRD peak area, which is from FAPbI₃ decomposition, was

chosen as the transformational parameter fraction and noted as $x(t)$. Following the study by Moore *et al.*[23] leads to kinetic equations:

$$\ln(t_{x2} - t_{x1}) = \frac{E_a}{RT} - \ln k_0 + \ln(\beta_{x2} - \beta_{x1}) \quad (1)$$

$$x(t) = 1 - \exp\left[-\left(k_0 \exp\left(\frac{-E_a}{RT}\right)t\right)^n\right] \quad (2)$$

where $t_{x1}$ and $t_{x2}$ are the time at which the transformed fraction is $x_1$ and $x_2$, $E_a$ is the effective activation energy, $R$ is the gas constant, $T$ is the temperature, $k_0$ represents the rate constant prefactor, $n$ is the growth constant and $\beta_{xn}$ is a state property that is independent of time/temperature variables. More detailed information regarding the derivations and explanation of the equations can be found in the study by Moore *et al.*[23] and kinetic modelling review by Liu *et al.*[34]. Figure 5a shows a fitted $x(t)$ for isothermally annealing at 210 °C (green, diamond), 230 °C (blue, triangle) and 250 °C (red, circle). Applying equation (1), Fig. 5b shows the extraction of effective activation energy of decomposition FAPbI₃, which is found to be $96.5 \pm 13.1 kJ mol^{-1}$. This may find use in lifetime predictions for FAPbI₃ solar cells. Fitting the Fig. 5a data into the JMA model, shown in equation (2), the kinetic reaction prefactor $k_0 = 1.82 \times 10^7 \pm 1.28 \times 10^5$ and the growth constant $n$ is close to 2 at 210 °C and approaches 3 at 250 °C. This growth constant dependence on the temperature suggests the growth of the decomposition product (PbI₂) is roughly three-dimensional at higher temperature and two-dimensional (2D) at lower temperature. Applying the JMA model to the halide perovskite decomposition system allows for determination of the FAPbI₃ decomposition activation energy and decomposition dimension, offering quantitative evaluation of the perovskite kinetics.

## Discussion

In this work we show that processing FAPbI₃ using RTA produces devices with comparable efficiencies to those made on a hotplate. We determined the processing phase space of FAPbI₃ with *in situ* XRD using an in-house-designed RTA and identified time and temperature ranges that can be used to produce good quality films; these are broader than the standard FAPbI₃ annealing conditions. The JMA kinetic model was applied to the halide perovskite decomposition process (into PbI₂), and the decomposition activation energy is determined, which will be useful in determining the inherent lifetime of FAPbI₃. This work is promising for the adaptability of industrial production of FAPbI₃, opening up the potential of

using more rapid ramp rates for processing with fast throughput, such as belt furnaces typically used in manufacturing.

## Methods

**Materials.** Unless stated otherwise, all materials and solvents were purchased from Sigma-Aldrich and used as received. FAI was from Dyesol. $PbI_2$ (99.999%) was purchased from Alfa Aesar. Spiro-OMeTAD (> 99.5%) was from Lumtec. Fluorine-doped $SnO_2$-coated transparent conducting glass (FTO) was purchased from Thin Film Devices Co.

**FAPbI$_3$ film deposition.** FAPbI$_3$ perovskite solution was prepared and deposited as reported by Wozny et al.[15], with slight modification. Specifically, a 0.7 M stoichiometric FAI and PbI$_2$ in anhydrous DMF solution were prepared, in a glovebox, and stirred for 2 h at room temperature. The resulted clear bright yellow FAPbI$_3$ solution was filtered with 0.20 µm polyvinylidene difluoride filter and spin coated, in a glovebox, on the compact TiO$_2$/FTO substrate by a consecutive three-step process: 500 r.p.m. for 3 s, 3,500 r.p.m. for 10 s and 5,000 r.p.m. for 30 s. One to 2 s before the end of second step, a drop of toluene was gently place on the spinning substrate to wash off the extra DMF solvent. The resultant transparent film was then placed in a jar, concealed tightly, took out of the glove box, took out of the jar and annealed in the RTA equipment (Ulvac MILA-3000 Minilamp Annealer). The transfer time from the spin coater to the RTA is typically 2–5 min. In the XRD experiments, the time between spin coater and RTA is roughly 10–20 min. Once the annealing is done, the annealed films were placed back into the jar and transferred to the N$_2$ glove box. For hotplate annealed films, the as spin-coated FAPbI$_3$ films were transferred on the hotplate after three minutes of waiting to match the RTA film transfer time.

**Solar cell fabrication.** Pre-patterned FTO glass slides (1 × 1 inch) were cleaned with deionized water and 2-propanol, and ultrasonic bathed in 2-propanal and acetone for 10 min each before performing a 15 min ultraviolet ozone cleaning. A thin (∼30 nm) compact TiO$_2$ was deposited on the FTO by spin coating 0.2 M titanium diisopropoxide dis(acetylacetonate) (Sigma-Aldrich, 75 wt% in iso-propanol) in 1-butanol (Sigma-Aldrich, 99.8%), with spin coater recipe as 700 r.p.m. for 10 s, 1,000 r.p.m. for 10 s and 2,000 r.p.m. for 30 s. The compact TiO$_2$/FTO substrate was then annealed at 500 °C for 1 h. Before depositing the perovskite photoactive layer, the substrates were cleaned with ultraviolet ozone for 15 min and transferred into a N$_2$ glove box where a spin coater (Laurell WS-650) is installed. The perovskite layer was deposited and processed as stated above. The hole transporting layer was deposited on the perovskite layer by spin coating, with spin recipe as 3,000 r.p.m. for 30 s, 70 µl Spiro-OMeTAD solution, which consisted of 72.3 mg of Spiro OMeTAD dissolved in 1 ml of chlorobenzene, 28.8 µl of 4-tert-butylpyridine and 17.5 µl of a bis(trifluoromethanesulfonyl)imide Lithium salt (Li-TFSI) solution. The Li-TFSI solution consisted of 520 mg of Li-TFSI dissolved in 1 ml of acetonitrile. The thickness of Spiro-OMeTAD layer was measure to be 100 nm. In the end, the gold electrode was thermally evaporated on the Spiro-OMeTAD /FAPbI3/compact TiO2/FTO glass with a thickness of 100 nm.

**Photovoltaic characterization.** Absorption measurements were taken on a Shimadzu UV–vis–NIR 3600 spectrometer at room temperature. SEM images were obtained using an FEI Nova NanoSEM 630. Solar cell devices were measured under AM1.5 illumination in a N$_2$ glovebox using a solar simulator (Newport, Oriel Sol3A) calibrated with an National Renewable Energy Laboratory-certified Si photodiode (Hamamtsu, S1787-04) equipped with an infrared-cutoff filter (KG3, Schott). A digital source meter (Keithley 2400) is used as external voltage to perform the current J–V characteristics. The devices were light soaked for 1–1.5 min before performing the J–V scan. The J–V scan rate is 100 mV s$^{-1}$ with reverse scan as 1.3 V to -0.2 V and forward scan as −0.2 V to 1.3 V. A metal aperture of 0.06 cm$^2$ was used when measuring J–V curves.

**In situ XRD data collection.** The in situ XRD was performed with the RTA chamber described in Ahmad et al.[29] using beamline 7-2 at SSRL with a photon energy of 12 or 12.6 keV (depending on the experiment). The sample temperature was measured with a sensitive thermocouple that was previously shown to be accurate by comparison of the temperature-dependent Ag diffraction data with the thermocouple[35]. Lead halide perovskites are beam sensitive and so to avoid significant beam damage, X-ray exposure was limited to 30 s for the duration of the anneal. The duration of 30 s was determined based on the observation that there was no visible evidence of X-ray damage after this exposure. Additional evidence that our data collection protocol did not result in beam damage came from comparing the XRD from a film exposed to the beam for 30 s to that from a part of the same film but not exposed to X-rays (for example, the sample was shifted) where there were no noticeable changes in the diffraction pattern. To limit the X-ray exposure to 30 s, 60 scans were taken with a 0.5 s exposure time and the X-ray beam was closed when the data was not being collected. To obtain data during the rapid conversion from the precursor to the perovskite as well as the longer conversion to PbI$_2$ and any grain growth, for longer anneal

times, 30 scans at a shorter interval between scans (1–2 s) was followed by 30 scans at a longer interval depending on the overall scan time (for instance, 30 s in the case of 15 min annealing). The RTA chamber was purged with N$_2$ for 5–10 min, after loading the as spin-coated films into the chamber, to reduce the O$_2$ and H$_2$O content of the annealing chamber. After purging, RTA from the RTA halogen lamps heated the samples, whereas XRD data were obtained while still flowing N$_2$. The annealing temperature profiles were such that they overshoot the set temperature by <5 °C and settled to the set temperature.

**XRD data analysis.** The diffraction was taken using a Pilatus 300k and the data is converted from 2D to one-dimensional (1D) using WxDiff, code written by Mannsfeld[36] to convert 2D data to 1D data. Once converted into 1D data, the data are loaded into IGOR Pro, and selected peaks are fitted using the Multi-peak fit with a Gaussian peak shape and a linear background. In this way the peak area, FWHM and associated error are extracted for the chosen peaks. The time and temperature that a phase appears (or disappears) are defined as the average (in time and temp) of the last scan that did not have the peak and the first scan that has the peak (or vice versa), with the error being half the difference. This is illustrated in Supplementary Fig. 8. The time reported in the time versus set temperature (Fig. 4) and temperature versus time plots (Supplementary Fig. 6) has been corrected such that the $t = 0$ is defined as the time the temperature reaches 30 °C, to avoid deviations (on the order of ≤1 s) in the temperature ramp due to differing initial temperatures.

**Data availability.** The data supporting the findings of this study are available from the corresponding author upon request.

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

## Acknowledgements

This work was funded by the Bridging Research Interactions through collaborating the Development Grants in Energy (BRIDGE) program under the SunShot initiative of the Department of Energy (DE-EE0005951). The work at the National Renewable Energy Laboratory was supported by the Department of Energy under Contract Number DE-AC36-08GO28308 and the work at Stanford Synchrotron Radiation Lightsource was funded by the Department of Energy under Contract No. DE-AC02-76SF00515. Stanford Synchrotron Radiation Lightsource at the SLAC National Accelerator Laboratory is a national user facility operated by Stanford University on behalf of the US Department of Energy, Office of Basic Energy Sciences. We thank Bart Johnson (SLAC) for assistance with SSRL beam line 7-2. We thank Dr Z. Li (NREL) for assisting on SEM images, Dr D. Moore (NREL) for discussion on the kinetic modeling and Dr J. Christians (NREL) for helpful discussions.

## Author contributions

V.L.P., B.D., M.F.A.M.v.H. and M.F.T. conceived and designed the study. B.D. fabricated and characterized all the devices. T.R.K.-S. assisted on device RTA. M.F.A.M.v.H. and B.D. prepared the films for the X-ray experiment. V.L.P., B.D. and M.F.A.M.v.H. conducted the X-ray experiments. V.L.P. analysed the XRD data. B.D. performed the kinetic modelling. M.F.A.M.v.H. and M.F.T. directed the project. B.D. and V.L.P. wrote the manuscript and all authors discussed, revised and approved the manuscript.

## Additional information

**Competing financial interests:** The authors declare no competing financial interests.

