## [Peer Review File · Nature Communications]

Reviewers' comments:

Reviewer #1 (Remarks to the Author):

Excellent paper, well written presenting interesting and new results.

Specific corrections:

Suggest including figure S2 from supplementary information in the main body of the manuscript and make figure 1a a separate figure, and combining S2 with 1b into a new figure 2.

Page 6, can the author comment on the spectral emission of the IR lamps in the furnace, is the energy in the visible and UV region of the spectrum excluded from irradiating the sample. Could this have any photo-catalytic effect on the reaction ?

Page 8, is there any difference in the film structure when characterized with in-situ XRD at the growth temperature, compared to measurements of XRD after growth at room temperature.

Page 12, suggest revision of figure 4 to contain part (a) and (b) with S6 as (a) and current Fig 4 as (b) so the different processing zones are clearly identified.

Can the author comment on the stress in the film after growth and how this varied with processing time and temperature.

Ref 24, the volume and page numbers are missing.

Ref 25 and 30 the journal title, volume and page numbers are missing.

Reviewer #2 (Remarks to the Author):

This work presents a method for annealing perovskite using IR radiation in a commercially available RTP system and deploys an in-situ XRD system in order to identify temperature dependent phase changes during the annealing process. This method for processing perovskite is not new and has been demonstrated previously (Rapid processing of perovskite solar cells in under 2.5 seconds, DOI: 10.1039/c5ta00568j) where a near infra-red halogen system is used to anneal perovskite in approximately 2 seconds with comparable performance to hot plate annealed samples, however this work is not noted in the paper. The second core experiment within the paper is the use of in-situ XRD analysis to determine the phase changes and although again not the first example of this method, it does provide a welcome insight into the evolution of process – performance parameters. In particular the use and layout of the processing phase space (Figure 4 and S7) has the potential to become a useful tool in determining annealing requirements and process windows during large scale fabrication. For this reason I recommend accepting the manuscript with a request that the authors consider the following comments:

(a) The authors should reconstruct the introduction in order to include prior developments in the area of (i) radiative annealing of perovskite (Rapid processing of perovskite solar cells in under 2.5 seconds, DOI: 10.1039/c5ta00568j) and (ii) The thermal gain in FTO glass under IR exposure due to free carrier absorption (Near Infrared Radiation as a Rapid Heating Technique for TiO₂ Films on Glass Mounted Dye-Sensitized Solar Cells, DOI: 10.1155/2014/953623).

(b) How is the sample temperature within the chamber accurately measured? Given the free carrier absorption of the FTO, how is the temperature at this critical interface determined and is it likely to differ from any convective air based thermocouple measurement?

(c) How scalable is the static chamber based RTP system?

(d) The lower stabilised photovoltaic performance should be included alongside the champion data in

the main manuscript or at least the variation noted in the text.

REVIEWERS' COMMENTS:

Reviewer #2 (Remarks to the Author):

Thank you for addressing my comments, I recommend that the manuscript be accepted.

Reviewers' comments:

Reviewer #1 (Remarks to the Author):

Excellent paper, well written presenting interesting and new results.

We thank the referee for this statement.

Specific corrections:

Suggest including figure S2 from supplementary information in the main body of the manuscript and make figure 1a a separate figure, and combining S2 with 1b into a new figure 2.

The original Fig. S2 is now combined with Fig. 1b, c, d and the original Fig. 1a is now Fig. S1. The original Fig. S1 is now Fig. S2. The text on page 6 ,7 has changed accordingly.

Page 6, can the author comment on the spectral emission of the IR lamps in the furnace, is the energy in the visible and UV region of the spectrum excluded from irradiating the sample? Could this have any photo-catalytic effect on the reaction?

The radiative thermal annealing chamber we used in this study used halogen lamps without any filter. As the halogen lamp spectrum shown below, a small part of the radiation spectrum is in the visible, however the fact that we do not see any difference between the FAPbI₃ films that are hotplate annealed and radiatively annealed suggests that there is no photo-catalytic effect on the reaction. We have added a statement to this effect in the main text on pages 6 and 7.

source: USHIO halogen lamps technical specifications.

Page 8, is there any difference in the film structure when characterized with in-situ XRD at the growth temperature, compared to measurements of XRD after growth at room temperature.

There are no noticeable changes in the diffraction pattern when the sample was shifted and measured on a part of the sample that was unexposed to the beam during the in-situ X-ray, as

commented on the page 18 of the paper. The XRD patterns of a spot which was exposed to 60 scans of X-ray and a spot with no prior beam exposure, measured at room temperature after annealing at 170°C for 15 min, are presented below.

Page 12, suggest revision of figure 4 to contain part (a) and (b) with S6 as (a) and current Fig 4 as (b) so the different processing zones are clearly identified.

Thank you a lot for the suggestion. However, with the diffraction data and average device performance, Fig. 4 already has a lot of information. So adding Fig. S6 to Fig. 4 will distract readers from the key information of Fig. 4. Also, even though measured temperature is an important term, the set temperature is more relevant in the context of the device efficiency.

Can the author comment on the stress in the film after growth and how this varied with processing time and temperature?

Both of the in-situ XRD and SEM characterization of the film suggest the grain size of FAPbI₃ is set once the film is deposited. Therefore, before FAPbI₃ degraded to PbI₂, there is no noticeable thermal induced stress on the film. In addition, comparison of the XRD spectrum of an annealed film cooled to room temperature with that of the trigonal room temperature standard from ref 28 suggests a strain of tensile strain of about 0.3%, but given the accuracy of our data (due to area detector), this is within the uncertainty of the measurement. Thus, we are reluctant to draw a conclusion from this result. But this is an interesting point, deserving of further work.

Ref 24, the volume and page numbers are missing. Ref 25 and 30 the journal title, volume and page numbers are missing.

Thanks for pointing this out. We have revised the references accordingly and the revised references are highlighted in yellow.

Reviewer #2 (Remarks to the Author):

This work presents a method for annealing perovskite using IR radiation in a commercially available RTP system and deploys an in-situ XRD system in order to identify temperature dependent phase changes during the annealing process. This method for processing perovskite is not new and has been demonstrated previously (Rapid processing of perovskite solar cells in under 2.5 seconds, DOI: 10.1039/c5ta00568j) where a near infra-red halogen system is used to anneal perovskite in approximately 2 seconds with comparable performance to hot plate annealed samples, however this work is not noted in the paper. The second core experiment within the paper is the use of in-situ XRD analysis to determine the phase changes and although again not the first example of this method, it does provide a welcome insight into the evolution of process – performance parameters. In particular, the use and layout of the processing phase space (Figure 4 and S7) has the potential to become a useful tool in determining annealing requirements and process windows during large scale fabrication. For this reason, I recommend accepting the manuscript with a request that the authors consider the following comments:

(a) The authors should reconstruct the introduction in order to include prior developments in the area of (i) radiative annealing of perovskite (Rapid processing of perovskite solar cells in under 2.5 seconds, DOI: 10.1039/c5ta00568j and (ii) The thermal gain in FTO glass under IR exposure due to free carrier absorption (Near Infrared Radiation as a Rapid Heating Technique for TiO₂ Films on Glass Mounted Dye-Sensitized Solar Cells, DOI: 10.1155/2014/953623).

Thanks for the good suggestion, which has been incorporated into the first paragraph of the page 4 as follows:

Beyond hotplate annealing, there are a few reported studies on using optical annealing approaches. Watson et al. reported using near-infrared radiation (halogen lamp)¹⁷ and photonic flashing (xenon lamp)¹⁸, and Druffel et al.¹⁹ proposed the use of intense pulsed light (xenon lamp) for sintering lead halide perovskites. The use of such optical annealing not only allows the sample to be effectively heated by absorption in the active layer, but also by absorption in the FTO substrate²⁰. However, the previous works only studied methylammonium based perovskite, such methylammonium lead iodide (MAPbI₃) and mixed halides such as MAPbI_{3-x}Cl_x, and the power conversion efficiencies are mostly not as good as those obtained by hotplate/oven annealing. Moreover, with such flash annealing techniques, it is not straightforward to control the temperature accurately and therefore these are not very well suited for conducting temperature related studies that are important in the perovskite field.

(b) How is the sample temperature within the chamber accurately measured? Given the free carrier absorption of the FTO, how is the temperature at this critical interface determined and is it likely to differ from any convective air based thermocouple measurement?

The sample temperature was measured with a sensitive thermocouple. This has been proven to be an effective and accurate way of measuring temperature as shown in our earlier studies (Fig.5, Fields et al., Nature Communications 7 (2016) DOI: 10.1038/ncomms11143) where temperature dependent Ag diffraction signal was compared to the measured (thermocouple) temperature from the RTA chamber. The temperature from thermocouple and the temperature determined with Ag diffraction are found to be very close at high ramp rates (100°C/s). For this study the maximum temperature and ramp rate are significantly lower, and therefore we believe the accuracy of the temperature measured by the thermocouple is even more precise. We have added a statement regarding this in the main text on the page 18.

(c) How scalable is the static chamber based RTP system?

The RTA or RTP system used in this study for making devices is MILA-3000 Minilamp Annealer which is only applicable for making 1 inch² devices, but it can be scaled up, as demonstrated in the semiconductor industry, which uses the technique to process large wafers (400 mm round). It can be envisioned that a system for even larger sizes can be engineered.

(d) The lower stabilized photovoltaic performance should be included alongside the champion data in the main manuscript or at least the variation noted in the text.

This suggestion is similar to the first comment from reviewer 1. The Stabilized photovoltaic performance is now included into Fig. 1 of the main text.